# AIS Data Vulnerability Indicated by a Spoofing Case-Study

Andrej Androjna [1,*], Marko Perkovič [1], Ivica Pavic [2] and Jakša Mišković [3]

[1] Faculty of Maritime Studies and Transport, University of Ljubljana, 6320 Portorož, Slovenia; marko.perkovic@fpp.uni-lj.si

[2] Faculty of Maritime Studies, University of Split, 21000 Split, Croatia; ipavic71@pfst.hr

[3] Croatian Defence Academy, University of Split, 21000 Split, Croatia; jmiskovic@unist.hr

[*] Correspondence: andrej.androjna@fpp.uni-lj.si

**Abstract:** This paper takes a close look at the landscape of the Automatic Identification System (AIS) as a major source of information for maritime situational awareness (MSA) and identifies its vulnerabilities and challenges for safe navigation and shipping. As an important subset of cyber threats affecting many maritime systems, the AIS is subject to problems of tampering and reliability; indeed, the messages received may be inadvertently false, jammed, or intentionally spoofed. A systematic literature review was conducted for this article, complemented by a case study of a specific spoofing event near Elba in December 2019, which confirmed that the typical maritime AIS could be easily spoofed and generate erroneous position information. This intentional spoofing has affected navigation in international waters and passage through territorial waters. The maritime industry is neither immune to cyberattacks nor fully prepared for the risks associated with the use of modern digital systems. Maintaining seaworthiness in the face of the impact of digital technologies requires a robust cybersecurity framework.

**Keywords:** Automatic Identification System (AIS); cybersecurity; anomaly detection; spoofing; maritime situational awareness; the safety of navigation

## 1. Introduction

Maritime industry stakeholders built the AIS with the goal of improving maritime safety, security, and MSA. The AIS was launched as a joint effort of the International Maritime Organization (IMO), the International Association of Marine Aids to Navigation and Lighthouse Authorities (IALA), the International Telecommunication Union (ITU), and the International Electrotechnical Commission (IEC) [1]. The IMO provides carriage requirements and performance standards. The IALA provides basic technical requirements, while the ITU and the IEC provide the broadcasting and electrotechnical standards of AIS. Although the development and deployment of AIS has contributed significantly to increased navigational safety, it has also resulted in numerous safety and security issues.

Originally, AIS was introduced on certain types of International Convention for Safety of Life at Sea (SOLAS) ships to assist the Officer of the Watch (OOW) in making decisions in the event of a collision. AIS provides the OOW with the most accurate and detailed data on all detected targets. This is very important for the OOW, especially targets that are at risk of or dangerously close to collision [2]. In addition, the data from AIS can be integrated into the Automatic Radar Plotting Aid (ARPA) and chart systems (Electronic Chart Display and Information System (ECDIS) or Electronic Chart Systems (ECS)) [2,3]. This provides the OOW with an additional source of navigation data (e.g., Closest Point of Approach (CPA)) that can help the OOW navigate or mislead him when the AIS station is spoofed. Although the use of AIS on ships has increased the safety of navigation, it is important to emphasize that AIS is not intended to guide navigation but to exchange navigational and other relevant data between ships and shore stations [4].

According to SOLAS regulation V/19.2.4, all ships of 300 gross tonnage (GT) and more engaged in international voyages and cargo ships of 500 GT and more not engaged

in international voyages, as well as passenger ships, irrespective of their size, shall be equipped with AIS [5]. Behind these international regulations, the European Union and national authorities have also developed obligations regarding AIS for non-SOLAS ships (i.e., fishing and pleasure boats).

To date, various AIS stations have been developed. According to IALA guideline 1082, AIS stations can be grouped by class and function into mobile and fixed stations with corresponding types of AIS. Mobile AIS stations are grouped by class into shipborne class A and B stations, AIS SART, MOB-AIS, EPIRB-AIS and AIS and SAR aircraft, while fixed AIS stations are AIS base stations (AIS BS), AIS repeaters, and AIS aids to navigation (AtoN) [1]. Figure 1 shows the classification of AIS stations.

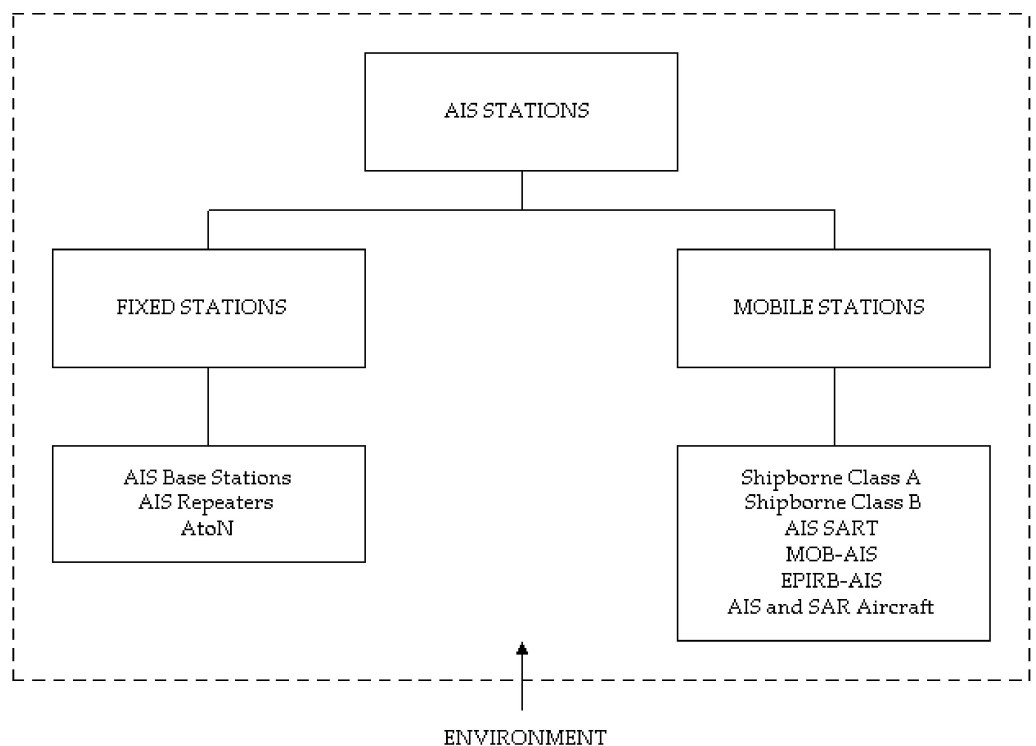

**Figure 1.** Types of AIS stations, according to the authors of [1,6].

Satellite-based AIS (S-AIS) have been introduced [7,8], and further development will include AIS used in the IMO e-navigation concept [6]. With the introduction of S-AIS, the coverage area of AIS has been extended from the very high frequency (VHF) range up to 400 km [9]. Although some countries already use AIS data from national satellites to monitor vessel traffic in sea areas off the national coast, global coverage of sea areas with S-AIS has not yet been established [7,8,10]. The wide use of this system and its advantages in the commercial sector will be recognized in the near future [11]. National authorities will be able to receive AIS signals from ships in the maritime zone under their jurisdiction directly from the AIS satellite service. In addition, data from S-AIS have a variety of regional and global applications as well as maritime spatial planning [8,12].

The transponders of AIS transmit and receive specific information on dedicated VHF frequencies [13]. The AIS automatically transmits and receives this information from similarly equipped ships, monitors and tracks ships, and exchanges data with land-based facilities [5]. The AIS uses the Time Division Multiple Access (TDMA) protocol, in which the time unit is divided into a number of equal slots containing a fixed amount of data, and the AIS station selects one or more free slots for data transmission [1,9,13,14]. The data is transmitted over a tracking system with a self-organized time division multiple access (SOTDMA) at a data rate of 9600 bits per second using Gaussian Minimum Shift Keying (GMSK) modulation. The ship transponder encodes the relevant information

in the AIS message string to within 256 bits. Messages are fragmented using National Maritime Electronic Association standard 0183 (NMEA0183) with a maximum sentence size of 82 characters; if an extended message is transmitted, the payload is split into multiple fragment sets, with a maximum of five (e.g., Binary addressed message or AIS application message 6 can be variable in length, up to a maximum of 1008 bits).

According to the amount of data to be transmitted, one or more time-slots can be required. The standard mandates that a transceiver can continuously transmit data for a maximum number of consecutive slots, as a continuous data stream, depending on its class. Class A devices can transmit continuously for five consecutive slots, while transmitting Class B devices can occupy the medium for a maximum of three consecutive slots, as per Table 1 [15].

**Table 1.** Maximum binary data bytes that could be transmitted using a given number of consecutive time-slots [15].

| Number of Consecutive Slots | Maximum Binary Data Bytes |
|:---:|:---:|
| 1 | 12 |
| 2 | 40 |
| 3 | 68 |
| 4 | 96 |
| 5 | 121 |

The key to the SOTDMA protocol is the availability of a highly accurate standard time reference (provided by the precise time signal in the Global Navigation Satellite System (GNSS) satellite transmission) to which all stations can synchronize their time slot assignments to avoid overlap [16]. The messages are transmitted with a power of 12.5 W (class A) and a bandwidth of 25 kHz. When a particular ship's message is broadcast, the station AIS announces its next broadcast position. The required ship reporting capacity under the IMO performance standard is at least 2000 time slots per minute at maximum 50% utilization. A single AIS channel capability is equal to 2250 messages transmitted within one minute, giving a slot length of 26.7 ms:

$$2250 \, \text{min}^{-1} = \frac{9600 \, \text{bit/s}}{256 \, \text{bit}} \cdot 60 \, \text{s} \tag{1}$$

AIS transmission can be on two channels, channel 87B on 161.975 MHz or 88B on 162.025 MHz, respectively, referred to as AIS channels 1 and 2, doubling the data capacity. The envisaged capacity of 4500 telegrams per minute is assumed to be sufficient for unrestricted ship-to-ship (2S) and even ship-to-shore (4S) use, with a typical range of 20 nautical miles (NM) between ships and up to 40 NM between ship and shore. The more accurate range (D) can be calculated using the heights ($h_1$, $h_2$) of two VHF antennas. The following example shows the signal range calculation for an AIS base station in Slovenia, located at Mount Slavnik at an altitude of 1028 m, successfully receiving data from the current vessel at a distance of over 90 NM. The height of the ship's VHF antenna is 30 m. The current reception data is also shown in Figure 2. Ships that are at the edge of the theoretical range of the AIS signal are detected, but no static ship data is available.

$$D = 2.5 \left( \sqrt{h_1} + \sqrt{h_2} \right) \tag{2}$$

$$D = 2.5 \left( \sqrt{1028 \, \text{m}} + \sqrt{30 \, \text{m}} \right) = 93.85 \, \text{NM} \tag{3}$$

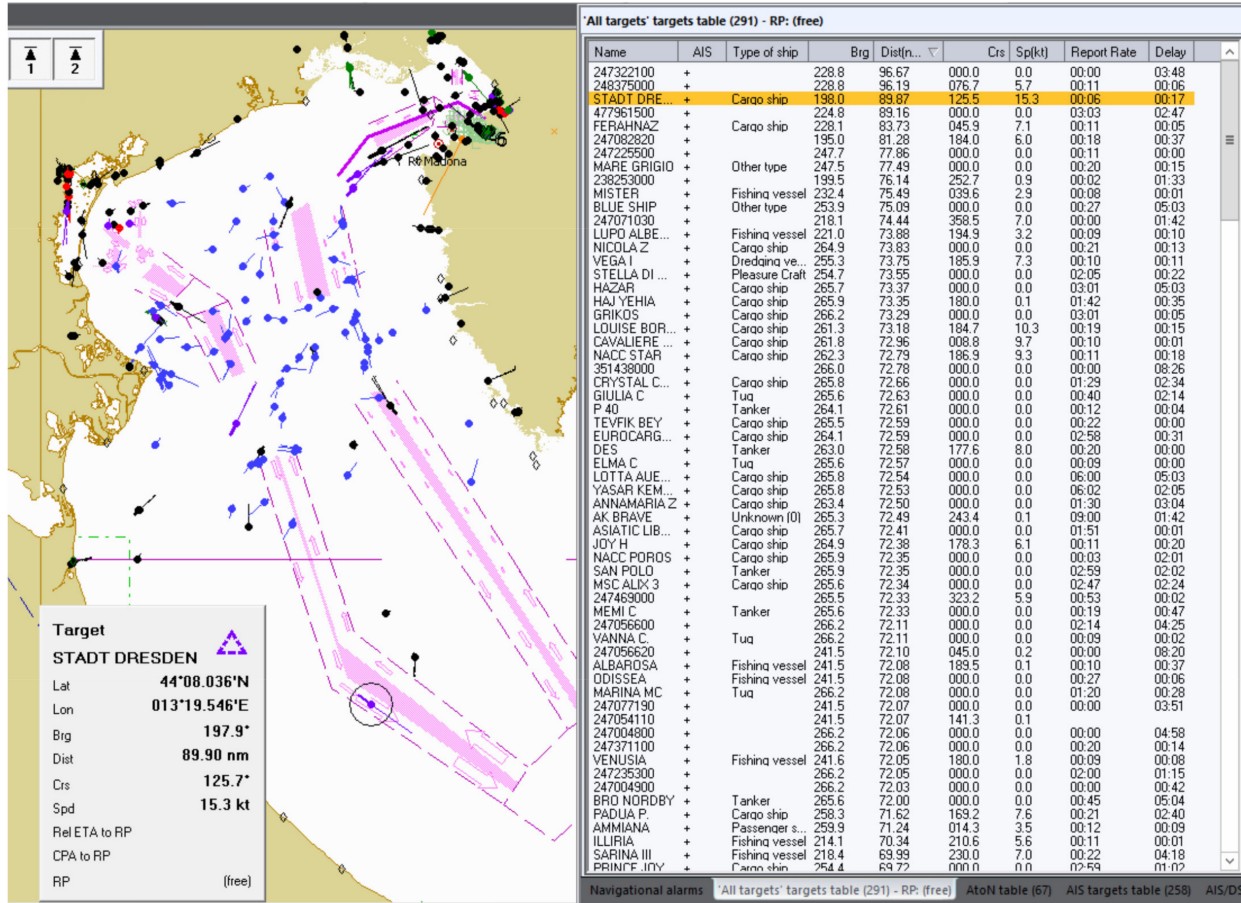

**Figure 2.** AIS target data (Screenshot of Navi-Harbor VTS application, Wärtsilä).

However, the detection probability of the AIS is not a purely geometric matter but depends on VHF propagation, which is affected by ground conductivity, atmospheric conditions, receiver sensitivity, antenna attenuation, signal shadowing, radio interference, and more.

The nominal reporting intervals for data transmission vary from 2 s to 6 min and depend on the type of AIS station, the group of messages, the navigational status, the speed, and the course change of ships [1]. Slower ships send kinematic data every 10 s, medium-speed ships every 6 s, high-speed ships every two seconds. If the ship changes heading, the transmission intensity increases by a factor of 3 (for slower and medium speed ships).

Table 2 shows the transmission intensity of static and dynamic information for Class A and B and for single and dual-channel transceivers. AIS transponders can receive all transmission information from both AIS channels simultaneously and combine the information from both channels into a single data stream. The standards of transmission, types, the format of messages, and symbols, make it simple for users to identify, monitor, and track targets detected by AIS.

**Table 2.** AIS reporting intervals [1].

| AIS Class A Transponder-Ships Dynamic Conditions | Dual-Channel | Single-Channel |
|---|---|---|
| Ship at anchor or moored | 3 min | 6 min |
| SOG 0–14 knots | 10 s | 20 s |
| SOG 0–14 knots and changing | 3.3 s | 6.6 s |
| SOG 14–23 knots | 6 s | 12 s |
| SOG 14–23 knots and changing course | 2 s | 4 s |
| SOG > 23 knots | 2 s | 4 s |
| Ship static information | 6 min | 12 min |
| **AIS Class B Transponder-Ships Dynamic Conditions** | **Dual-Channel** | **Single-Channel** |
| SOG < 2 knots | 3 min | 6 min |
| SOG > 2 knots | 30 s | 1 min |
| SOG | | |
| Ship static information | 6 min | 12 min |

There is additional AIS traffic present; AIS BS reports, AtoN, Application-Specific Messages and AIS-SART, MOB-AIS or EPIRB-AIS, all with variable reporting rates. In addition to the nominal reporting frequency, it is possible to poll ships for current information. A competent authority can use a base station to prompt mobile AIS devices for more frequent messages, such as vessels at anchor exposed to heavy wind [1]. If the altitude of the base station AIS is as shown in Figure 2 (1028 m) and it is located in an area with moderate shipping, and a large number of ports, including offshore industry, fishing, and yachting, VHF data link (VDL) congestion is common. When spoofing occurs in such an area, nautical safety can be affected.

Today's applications of AIS data has shifted from use for collision avoidance, identification, and tracking to monitoring shipping routes, maritime traffic trends, risk analysis, marine accident investigations, near-miss investigations, search and rescue operations, waterway planning, management and maintenance using AtoNs, traffic simulation and forecasting, fisheries monitoring, ecological matters, and prevention of illegal activities at sea (e.g., illegal fishing and prevention of piracy) [1,8,16–21]. The applications of S-AIS data will be even broader in the areas of marine environmental protection, maritime safety, and security [22].

This paper takes a closer look at the landscape of AIS as an essential source of information for MSA and identifies its vulnerabilities and challenges for safe navigation and shipping with a particular focus on spoofing. A case study of a specific spoofing event near Elba in December 2019 confirms that typical maritime AIS can be easily spoofed and generate erroneous position information. In conclusion, the implementation of a robust cybersecurity framework to improve defenses against spoofing and other maritime cyber threats is recommended.

The paper is structured as follows: Section 2 describes the methodology, Section 3 presents AIS vulnerabilities and practical examples of significant spoofing incidents, and Section 4 refers to a practical investigation of an AIS spoofing event. Section 5 presents the discussion, and Section 6 the conclusions.

## 2. Methodology

A systematic literature review followed by a comprehensive, explicit, reproducible and idiosyncratic implicit method of data collection was conducted and structured following the documented guidelines [23–25]. This method consists of ten steps that can be divided into three main phases. In the first phase, planning focused on defining a review question to guide the search: "What are the implications of AIS spoofing in the maritime domain?" The second (review) phase identified the appropriate time frame for documents to be included from relevant research databases such as Scopus, Science Direct, Web of Science, Google Scholar and open sources. "AIS" and "spoofing" were identified as search keywords to be reviewed. After refining the selection to identify relevant documents, they were analyzed

and synthesized by context, methodological approaches, and results. Our final list included 55 documents limited to AIS spoofing (28 articles, 18 peer-reviewed journal articles, and 9 reports from specialized agencies) covering the area of "AIS spoofing" in the period from 2019 to 2020. Some previous studies on this topic were also considered.

In the third phase (Reporting and Dissemination), we report our findings from the literature review. The specific aspect of AIS/Global Positioning System (GPS) spoofing is corroborated by the case study analysis in Section 4 from the Faculty of Maritime Studies and Transport, University of Ljubljana, regarding a particular AIS spoofing event near Elba Island in late 2019.

As part of the research, AIS data were first obtained through cooperation with the Slovenian Maritime Administration, which is stored at MARES regional data exchange program. Additional AIS day data were subsequently obtained from the Italian Coastguard. These data were used to analyze the strength of signals received at the AIS BS on the island of Elba. The French agency maintaining the AIS system and the commercial providers MarineTraffic and VesselFinder were also contacted. An investigation into the consequences of this event followed; seven vessels were found to be in a spoofing cloud, a letter was sent to those in charge of the International Safety Management code (ISM) implementation to enquire about the possible consequences for the safety of shipping. Afterward, the archive data were streamed again using the application AIS Network Data Client and played in two different VTS applications Navi-Harbor (Wärtsilä) and Pelagus (Elman). A navigation scenario was then configured in the affected area. The AIS spoofing data were displayed on ECDIS and RADAR applications via the ship tracking simulator Navi Trainer Pro (Wärtsilä). An analysis of the collected data follows. Finally, using the European Marine Observation and Data Network (EMODnet) method [26], a traffic density map (TDM) was created by using ship positioning data from terrestrial and S-AIS data sources, maritime infrastructure, and the SafeSeaNet Ecosystem Graphical Interface (SEG) application.

## 3. Results

This chapter highlights the unique AIS challenges associated with problems in securing ships at sea. It presents findings on AIS vulnerabilities, some of the recent examples of AIS spoofing trajectories, and practical investigations of an AIS spoofing event. To understand the current research, it is important to understand the background, how AIS works, the liabilities, and the methods by which vulnerabilities to the system can be initiated. The nature of the cyber arena is changing today, and the maritime sector is experiencing sophisticated and complex attacks that no one could have predicted years ago.

### 3.1. AIS Vulnerabilities

To date, some studies have been conducted on the vulnerabilities of AIS. Balduzzi et al. [27] discussed the security evaluation of AIS and divided the AIS threats into three macro-categories: spoofing, hijacking, and availability disruption, based on software or radio frequency (RF) threats. Kessler pointed out that the vulnerability of the AIS VHF broadcast frequency can lead to bandwidth usurpation by potential attackers who can prevent other devices from transmitting, negatively affect the synchronization process, or change slot reservation/assignment information [28]. Goudosis and Katskias [16], as well as Aziz et al. [9], emphasized that AIS-broadcasted messages are neither encrypted nor authenticated, making them vulnerable to unauthorized manipulations.

The vulnerabilities of the AIS arise mainly from its technical performance. The AIS is an open-source system that transmits on VHF channels in the maritime mobile band [1,3]. The data sent by the transponders are simply not subject to any control or verification of their integrity [16,29]. Data exchange between stations in the AIS service is largely based on transmission and reception at RF, and therefore the AIS service is vulnerable to malicious transmissions and risks of data manipulation [1]. Figure 3 shows the concept of AIS service and possible vulnerabilities in the form of spoofing via RF. At the center of the AIS service are base stations designed for use by government agencies to manage the VDL to provide

wide Vessel Traffic System (VTS) or Coastal Surveillance coverage [1]. Base stations are vulnerable to possible attack if someone sends false AIS signals that can overload time slots and prevent or complicate the operation of regular AIS stations. AIS SAR devices are MOB-AIS, EPIRB-AIS, and AIS SART stations. These stations use a radio beacon system and are designed to transmit distress signals with geographic coordinates, allowing SAR units to more accurately locate survivors [27]. In addition to this advantage, if the attackers generate false distress signals, this could lead to the deception of shipborne AIS stations, VTS SAR services, and facilities. AtoN is also part of the AIS service that could enhance the safety of navigation, especially in coastal navigation (i.e., shallow waters, port approaches, navigational obstacles) with high traffic density. It should be emphasized that not all ships are equipped with AIS and ECDIS [30].

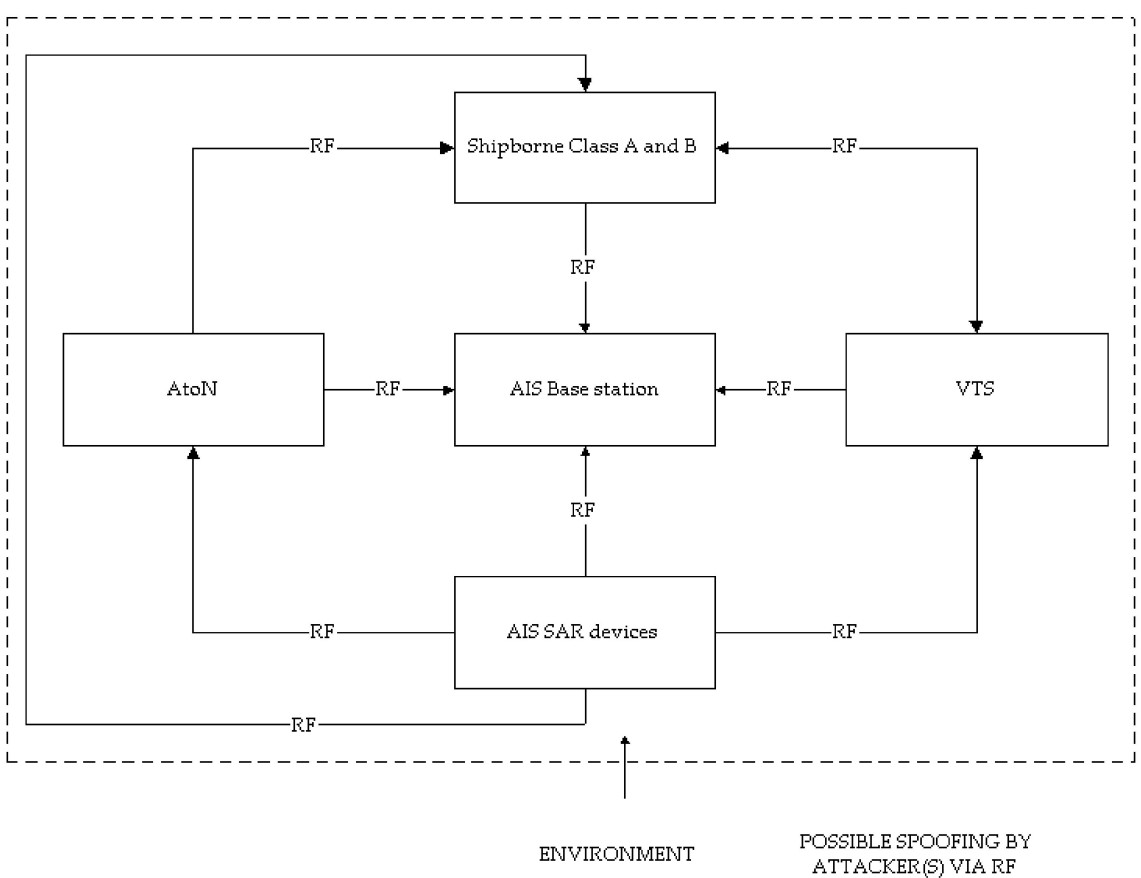

**Figure 3.** AIS mobile and fixed station RF-based spoofing, according to the authors of [3,6,27].

From an operational point of view, AIS AtoNs are identified as physical, synthetic and virtual [1]. Based on the technical characteristics, it is obvious that synthetic and virtual AtoNs (V-AtoN) are more vulnerable to spoofing than physical AtoNs. The V-AtoN can easily be spoofed intentionally [31]. Spoofed AtoN signals can mislead the OOW during navigation. Combining virtual and physical AtoNs is considered an effective measure to avoid the negative consequences of V-AtoN spoofing [32]. The possibility of V-AtoN spoofing should be considered as the main vulnerability due to the dependence of AIS on GNSS, which is also vulnerable to spoofing [30,33].

Ship-borne Class A and B as the main component of the AIS service could enhance MSA and collision avoidance. AIS vulnerabilities can result in different errors and falsifications of messages, which may have, in some cases, a significant impact on mariners and local authorities MSA. In case of a malicious attack, a false maritime picture could occur

and mislead the OOW in navigation or VTS. These may affect decision-making processes in navigation and surveillance.

Iphar et.al. [34] presented a risk analysis of the AIS supported by the EBIOS (fra. Expression des Besoins et Identification des Objectifs de Sécurité) methodology and an anomaly typology. The EBIOS method is used in risk evaluation and contributes to the security assessment of the AIS system. It consists of a 5-module procedure: Context study, study of dread events, study of the threat scenarios, study of the risks, study of the security measures. Following on from the EBIOS risk analyses, Iphar [34,35] presented different dread and threats scenarios and risks. Using the EBIOS method, 11 risks with an intolerable level linked to the AIS use have been identified. These risks are linked to integrity and availability: dynamic data, AIS data, transmission function, reception function, positioning and visualization function.

Some of the identified risks deal with data integrity issues, which is one of the data quality dimensions. To conduct AIS data quality assessment, different kinds of possible anomalies in AIS messages have been recognized and presented in Table 3.

**Table 3.** AIS anomalies, according to Iphar et.al. [34,35].

| Ais Anomalies | | | |
|---|---|---|---|
| Behaviour | Content of the Message | Lawfulness | Quality |
| Kinematic (position, move) data | Static data | Criminal issues | Unexpectedly changing data |
| Voyage data | Human data | Breach-level issues | Impossible data |
| | | | Impossible with respect to others information |
| | | | Missing data |

To assess AIS data and detect AIS anomalies, different algorithms and filters could be used in order to provide the maritime surveillance community with information on unusual behavior [36]. To prevent the AIS security weaknesses, some solutions have been proposed.

Kessler [28] proposed the protected AIS (pAIS) software, which uses a public-key cryptography method. The pAIS software provides authentication of the AIS message senders as well as a message integrity service. This method could be implemented by a simple software upgrade at the existing AIS protocol.

Aziz et al. [9] proposed the secure AIS application protocol, which is based on a pairwise key and aims to encrypt and authenticate the transmission of messages between two AIS stations using a certification mechanism. This application will be integrated in AIS Class A and B stations. Sciancalepore et al. [15] extended this work and introduced the Auth-AIS as an authentication scheme using the Deterministic Security Configuration model based on Timed Efficient Stream Loss-tolerant Authentication (TESLA) symmetric authentication protocol or Probabilistic Security Configuration model based on the coupling of TESLA with Bloom Filter tool. Those proposals require software updates and provides authentication of AIS messages.

Goudossis and Katsikas [16] proposed the architecture of maritime certificate-less identity-based public-key cryptography (mIBC) and discussed the feasibility of the implementation aspects of mIBC. The architecture will enable the use of extant AIS and an additional four secure modes of operation. This proposal provides on-demand authentication, message integrity, and confidentiality of AIS data. The implementation of this proposal requires some changes in the existing AIS service [16,37].

All these proposals are based on cryptography methods enabling encryption of AIS messages. They are also backward compatible, enabling interoperability with the existing AIS devices, which do not use the modified software or hardware.

The next section will provide analyses of the AIS spoofing incidents in recent years.

### 3.2. An Overview of AIS Spoofing Incidents in the Last Years

The last two years of the decade have been remarkable in many ways. While the coronavirus pandemic has disrupted operations worldwide, some global trends have continued unabated. One of these is our increasing reliance on the GNSS [38,39]. In recent years, there have been several disruptive incidents that have caused a stir in the shipping industry, as shown in Table 4.

**Table 4.** An overview of some GNSS spoofing events that affected maritime traffic between the years 2008 and 2020.

| Location and Date | Spoofing Incidents Description |
|---|---|
| The Southern Ocean, 2008–2018 | To disguise her illegal fishing operations, m/v Andrej Longov/Sea Breez 1/Ayda/STS-50 committed identity fraud by repeatedly falsifying her registry, producing multiple fake signals, and appearing in nearly 100 different locations simultaneously. |
| Gulf of Oman/Malaysia, September 2013 | M/v Ramtin was involved in "spoofing" by falsely transmitting her AIS identity during suspicious activities and deceiving authorities at Karachi port under the name of m/v Hamoda. |
| Ten global locations connected to one of the superpower states, 2016–2019 | 9883 suspected spoofing incidents. |
| The Black Sea, June 2017 | Vessel tracking systems placed many vessels near Novorossiysk Commercial Sea Port in the nonsensical location, on the Gelendzhik Airport. |
| The East China Sea, 28 October 2018 | M/v Yuk Tung was involved in "spoofing" by falsely transmitting its AIS identity in a suspicious ship-to-ship transfer and deceiving authorities under the name of m/v Hika, which was anchored in the Gulf of Guinea, more than 7000 m away. |
| Point Reyes in northern California, August 2018–June 2019 | Ships thousands of miles at sea mysteriously reported GPS positions in ring patterns off the coast of San Francisco. |
| Eastern Mediterranean and the Red Sea, 2018–2019 | Signal interference, loss of erratic AIS/GPS signals. |
| Strait of Hormuz, July 2019 | A British oil tanker, the Stena Impero, was seized by Iranian forces after the ship was spoofed into changing course into Iranian waters. |
| Ningbo (China)-Nampo (Democratic People's Republic of Korea), July–November 2019 | The m/v Fu Xing 12 manipulated its identity by employing two AIS on board and using four different ship names to disguise its operations in delivering illegal coal and other resources. |
| Port of Shanghai, 2018–2019 | Fake signals caused ships to appear to be moving in ring patterns at short intervals. |
| Ponce De Leon Inlet, Florida, 2020 | Four visual AtoNs appeared on the map based on fake AIS messages. |
| Elba Island, 3 December 2019 | Deliberate spoofing of the vast number of artificial AIS targets temporarily affected the navigation of ships. |
| Galapagos, July 2020 | One of the world's largest fleets of fishing nations misreported its location (approximately 10,000 km from its observed location) to conceal illegal fishing activities in the exclusive economic zone (EEZ) around the Galápagos Islands. |

From Table 4, it can be concluded that the original purpose of AIS spoofing to this day is to disguise illegal fishing and other illegal activities at sea, which includes ship spoofing and AIS hijacking. There have also been some examples of AtoNs spoofing (Ponce De Le-on Inlet). In recent years, we have seen GNSS spoofing as part of defense development in a civilian scenario. The AIS spoofing has been deliberately used for electronic warfare and concealment of military activities, just like the situation in the eastern Mediterranean and the Red Sea. The incident in the Black Sea led to speculation that it could be attributed to one of the tests of satellite spoofing technology by one of the space superpower states. Whether as part of their electronic warfare arsenal or simply as an anti-drone measure to protect very important persons [40]. A practical example of geopolitical and geo-economic competition is the July 2019 detention of a British oil tanker, the Stena Impero, which was seized by Iranian forces after she was tricked (spoofed) into altering its course into Iranian waters. As a result, the ship, her cargo, and crew had become more than pawns in a geopolitical war [41]. Many of the shipping companies operating in the region also instructed their ships to transit Hormuz only at high speed and during daylight. Nevertheless, we should not forget that one-third of the world's oil flows by sea, about 17 million barrels per day, through the strait, making it one of the most important oil trade routes in the world [40,41].



At the same time, a mysterious new electronic device has emerged in China that spoofs AIS signals in ways experts have never seen before. There have been reports of multiple spoofing incidents detected in over 20 coastal areas and ports (including Shanghai, Fuzhou (Huilutou), Qingdao, Quanzhou (Shiyucun), Dalian, and Tianjin) that have been ongoing for months. Unlike with 'traditional' spoofing, the GNSS signals were congregated into large circles, later dubbed "crop circles", moving signals shifted to the same position, resulting in a confusing traffic situation for the pilots of ships [42]. Bergman [43] observed that the locations of the "spoofing circles" were oil terminals. Spoofing timing, the imposition of sanctions on the purchase of Iranian oil by the United States, and observations by others that Iranian oil is entering China suggest that some spoofing is used to disguise these transactions. Circulation phenomena have also been observed in Tehran, Iran [44]. In all these cases, the real location was relatively close to the fake and circling locations. In some other cases observed in 2019, the real locations of the ships were thousands of miles away, literally on the other side of the globe. The approximate true location of the ships was verified by examining the field of view of the satellite that received the position reports from AIS. A satellite that could only see an area about 5000 km wide, for example, the Gulf of Guinea in West Africa, received GPS-based position data from a ship AIS, showing the ship off Point Reyes in northern California. Their position reporting also confirmed the true positions of the ships before and after the "displacement events". However, it is still unclear whether these errors were caused by the ships' AIS or by an error or interference with the GPS navigation receivers. In the previously observed cases, it seemed obvious that a jamming device was nearby and affecting many ships. In these incidents, each ship was the only one in its vicinity, and the ships were thousands of miles apart [44].

Another example of spoofing was observed in July 2020, when one of the world's top fishing fleets was accused of misreporting its location to conceal illegal fishing activities in the EEZ around the Galápagos Islands. The ships reported a location on New Zealand via AIS that was about 10,000 km away from their observed location, when in fact, they were near the Galápagos Islands, where illegal fishing has occurred before. The HawkEye[360] satellite system, which provides a unique method for obtaining maritime awareness through RF geospatial intelligence, revealed many instances of suspicious vessel behavior, including evidence that many vessels in one of the world's top fishing fleets stopped transmitting AIS for eight or more hours and may have penetrated deep into the Galápagos EEZ, as shown in Figure 4 [38,45].

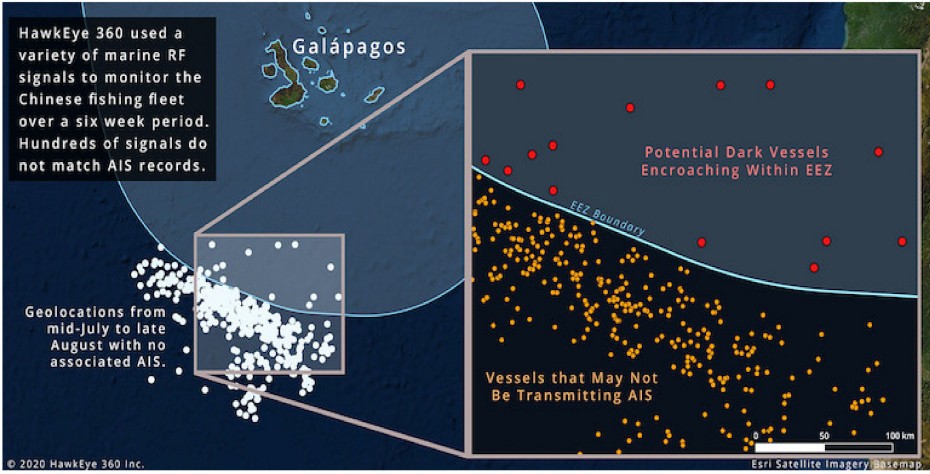

**Figure 4.** Vessels in red color were disappearing from AIS tracking, the Galápagos Islands [45].

This type of AIS "cloaking" is just one of many ways criminals use location spoofing of a GNSS-dependent system to support their nefarious activities [38].

However, the question of the manipulation of AtoNs in Florida and the spoofing of the AIS system observed in late 2019 near Elba Island remains open. An investigation into the latter is presented in the next section.

## 4. Case Study on Spoofing Large Amounts of AIS Fake Targets Nearby Elba Island and Its Impact on VDL

With the increasing demand for maritime VHF data communications, AIS is heavily used for maritime security, maritime situational awareness and port security [46]. VDL exposure remains an increasingly serious problem in many parts of the world due to the proliferation of AIS applications, message types, services, and equipment types, as well as unanticipated increases in user traffic.

The critical threshold of 50% VDL has already been exceeded in some areas of the world where shipping is at a high level. However, when intentional emission of virtual targets occurs in addition to the high AIS traffic, such actions can further jeopardize the operation of the system. The IALA has indicated that additional AIS channels are needed to provide maritime safety information and general data communications and to protect the integrity of the AIS VDL [46].

Each ship transmits its AIS data and receives the AIS data from all ships within its VHF radio range. In high traffic density situations, it becomes necessary to reduce the number of vessels in a communication cell. When the number of AIS messages begins to overload the network, the AIS station can automatically reduce the size of its cell by ignoring weaker, more distant stations in favor of nearby ones. When the data link is congested to the point that the transmission of safety information is compromised, two different methods are available to resolve the congestion [47–49]. First, the own station can intentionally reuse slots from the most distant station(s) within the candidate slot selection interval, but the slots allocated by the base stations can only be used if the base station is more than 120 NM away from the own station [48]. A distant station that was the subject of the intentional slot reuse is excluded from further intentional slot reuse for a period of one frame. This allows time slots used by weak stations far away to be used by stations nearby. This method is known as the Robin Hood principle [50].

In addition, to resolve congestion and protect VDL functionality, the base station may allocate Class A reporting rate to all non-Class A shipborne mobile stations, and to resolve congestion for Class A shipborne mobile stations, the base stations may use slot allocations to redirect slots to reserved Fixed Access Time Division Multiple Access (FATDMA) slots. In this way, the broadcasting system can be overloaded by up to 500% by sharing slots and still provide close to 100% throughput for ships closer than 8 to 10 NM to each other in a ship-to-ship mode: i.e., only more distant targets are affected by the outage in order to give priority to closer targets that are of greater interest to navigators and VTS operators. In practice, the capacity of the system is almost unlimited so that a large number of ships can be accommodated simultaneously.

The only known AIS spoofing case where a large number of fake AIS messages were delivered occurred on 3 December 2019, starting at 13:13 UTC, in the maritime area between the island of Elba and Corsica. Thousands of AIS streams were received and recognized as Dutch-flagged naval units that were artificially generated and had different identification codes, positions, routes, and speeds. This information completely saturated a circular sea area with a radius of about 11 NM. The attack was repeated three times, at 13:13 UTC (duration 3 min), at 13:28 (duration 4 min), and the last burst started at 13:37, lasting only a few seconds. From Figure 5 the generation of regular position reports with increasing Maritime Mobile Service Identity (MMSI) numbers is evident.

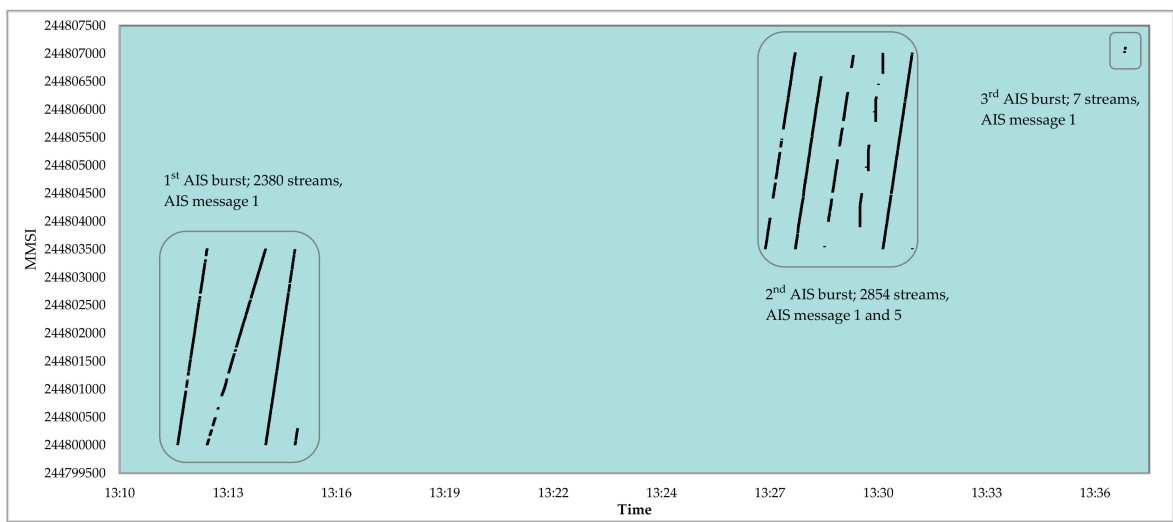

**Figure 5.** AIS bursting—MMSI selection and temporal occurrence.

During the first burst, 2389 messages were received, all messages were categorized as AIS message number 1. The smallest MMSI number was 244800009, and the largest was 244803507. In the second burst, the smallest MMSI received was 244803510, and the last was 244807010. We can assume that the spoofer transmitted all consecutive MMSI numbers, namely 3500 in the first broadcast and the same number in the second broadcast. The third broadcast was aborted immediately after it started for some inexplicable reason. A total of 5241 AIS messages or 3741 different MMSIs (ships) were received. During the second transmission, a large number of AIS messages number 5 were also received, also containing the identification, dimensions and type of the ship. All ships were identified as passenger ships (AIS type 60) with a length of 90 m and a width of 24 m, without draft and destination information. The names and call signs of the vessels were generic, namely VESS0 and CAL, adding the last three digits of MMSI (starting from 000 up to 999).

The following figures (Figures 6 and 7) show the spatial distribution of received AIS messages (for the 1st and 2nd burst), depicting real targets in the area, a cloud of artificial targets, and marked spoofed targets of particular interest. Right at the beginning of the first spoofing, two particularly interesting AIS targets also appear at the edge of the generated cloud of artificial targets. The first with the invalid MMSI number 999999999, which is quite common around AIS and often associated with navy vessels. The second vessel is a Belize-registered bulk carrier (MMSI 312320000) that has been scrapped since 2016. Individual fishing vessels are known to take on the identity of scrapped vessels.

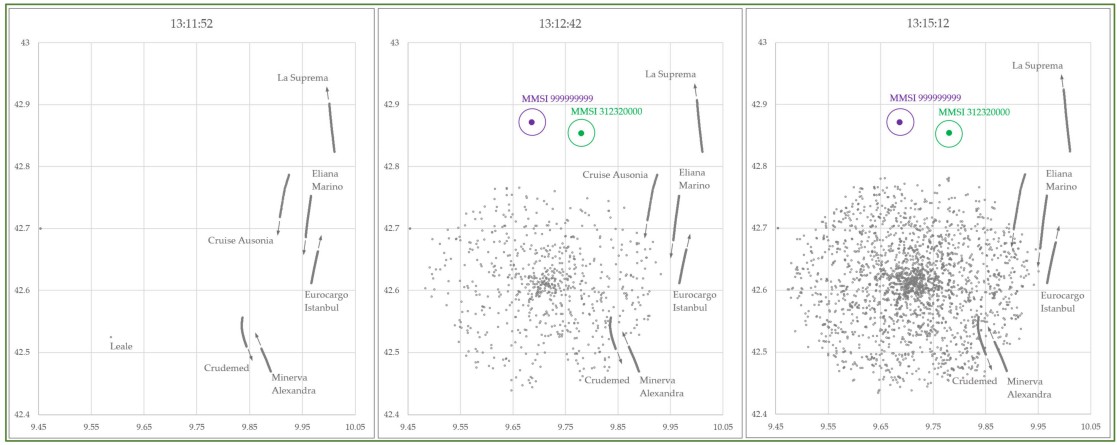

**Figure 6.** Spatial and temporal distribution of the received AIS signals during the 1st burst.

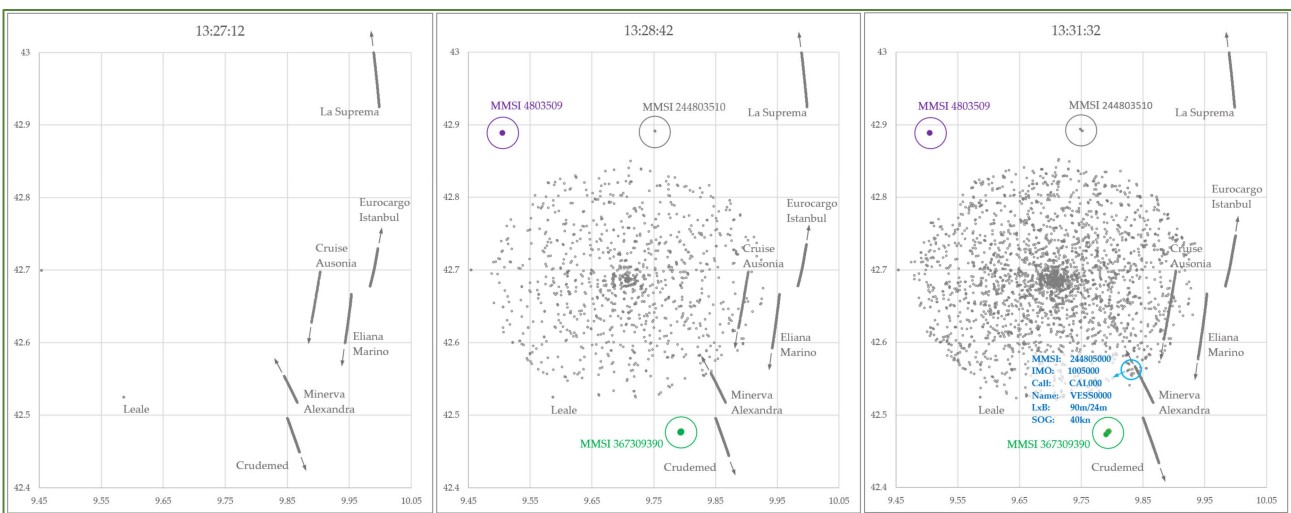

**Figure 7.** Spatial and temporal distribution of the received AIS signals during the 2nd burst.

Also, right at the beginning of the second spoofing, particularly interesting ships appear at the edge of the generated cloud. This time, one of the generic artificial ships emerges from the cloud and appears very close to the aforementioned ships from the first burst. In addition, one of the ships appears as a coastal station with the invalid MMSI number 4803509, which was generated by truncating the record from the generic MMSI data of the artificial ship's MID 24480**** by the first two digits. Thus one of the AIS targets, probably 244803509, is converted to an AIS BS signal.

Among other bursts, the US-registered vessel MMSI 367309390 also appears on the south side. This vessel is also fake. Among the many artificial AIS targets, the vessel MMSI 244805000 stands out; the number of a fishing vessel that has been spoofed several times in the past, this vessel is placed directly in front of the real vessel Crudemed. This vessel was also transmitting static data, like many others among the second burst. The speed of the ship is 40 knots. Figure 7 shows the position of this ship and its data with blue color text.

### 4.1. VDL Load and Spoofing Source Analysis

In this part, we must first point out that the ITCG has provided us with received data from other vessels (VDM), which shows that most of the AIS messages are received via the Monte Capanne AIS BS located on Elba Island. From the database, we can deduce that only 20 messages sent by artificial ships were received by the AIS BS in Corsica. At the same time, we have analyzed the signals of the real ships, almost all of which were received from another AIS BS, such as Monte Agentario or La Castellana. In Figure 8, the range of MMSI numbers used in spoofing was shown along with the number of repetitions (1 to 4) of each MMSI. From the same figure, the missing MMSIs can be seen. Received data indicates that a spoofing algorithm was run with auto-incrementing MMSI numbers. Missing MMSIs in received data sets are likely due to transmission collisions with nearby vessels.

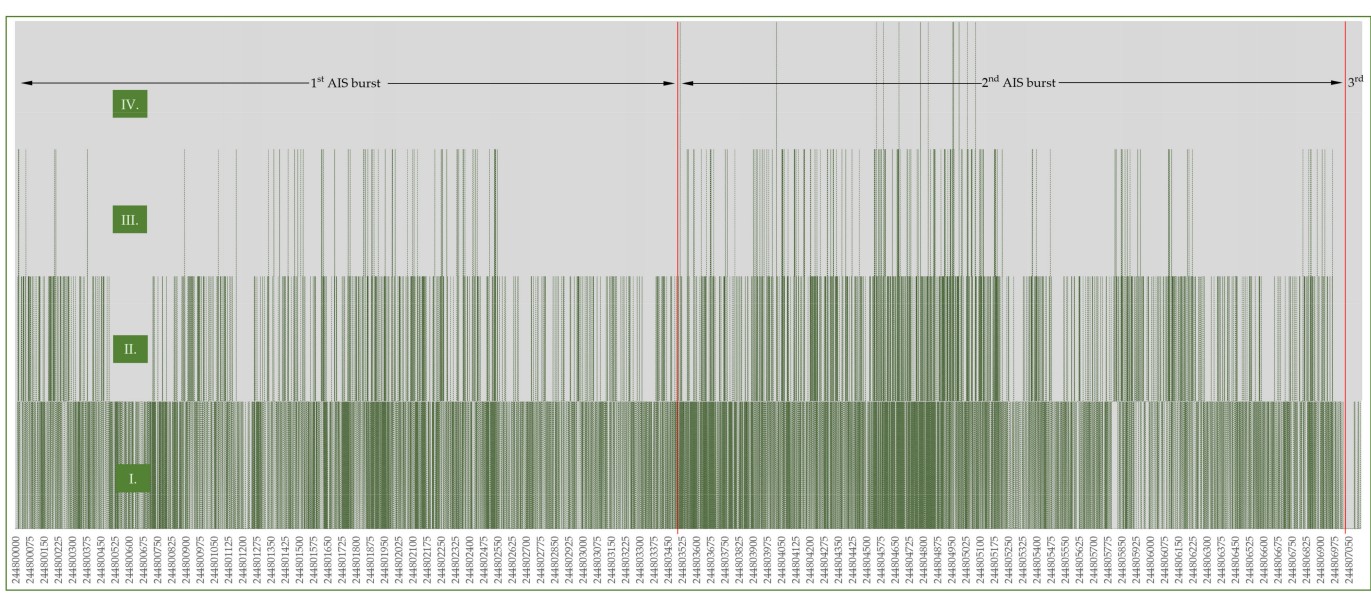

**Figure 8.** AIS bursting—MMSI temporal occurrence and repetition.

Figure 9 shows the distribution of the spoofed AIS targets during the second burst; the first part shows the principal component analysis with two principal axes of the distribution. The ship positions are shown in the Universal Transverse Mercator coordinate system. The second part of the figure shows the distribution of ships along each axis. It can be seen that it is most likely a parabolic spatial distribution, while the velocities of the ships and their courses are normally distributed. The Formula (4) show the most likely distribution model used. However, the right part of the figure also shows the Gaussian curves for the distribution along the primary axis for both bursts. It can be seen that the sources are shifted by about 5 km but have the same standard deviation of 7.7 km.

$$R_{max} = 11NM$$

$$\alpha = 360° \cdot RAND(r); 0 < r < 1$$

$$R = R_{max} \left( 1 - \sqrt{1 - RAND(r)} \right) \quad (4)$$

$$x = R \cdot \cos(\alpha); y = R \cdot \sin(\alpha)$$

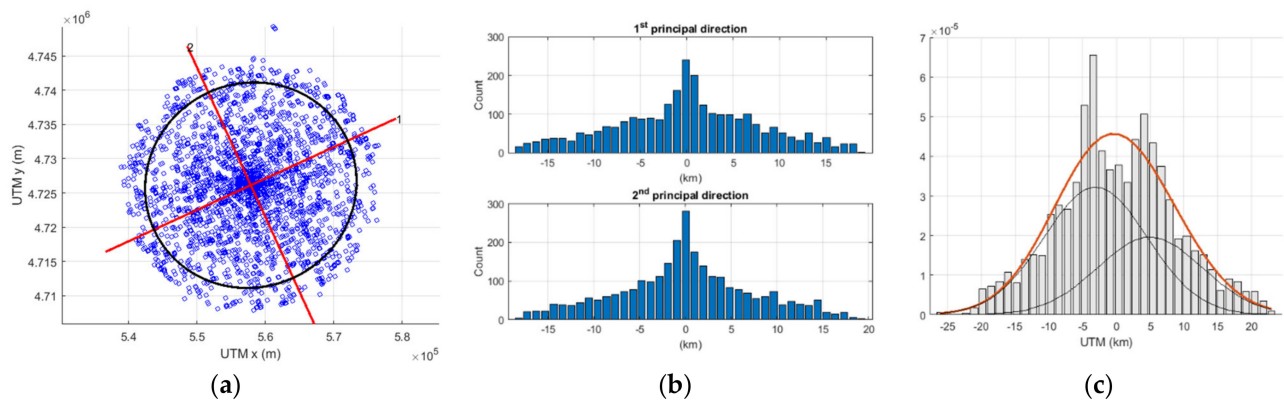

**Figure 9.** AIS bursting—(**a**) PCA analysis; (**b**,**c**) "Normal Mixture" distribution.

The manipulation of the AIS system is also illustrated in Figure 10. The EMODnet method showed a shipping density of up to 45 ships/km$^2$ [26].

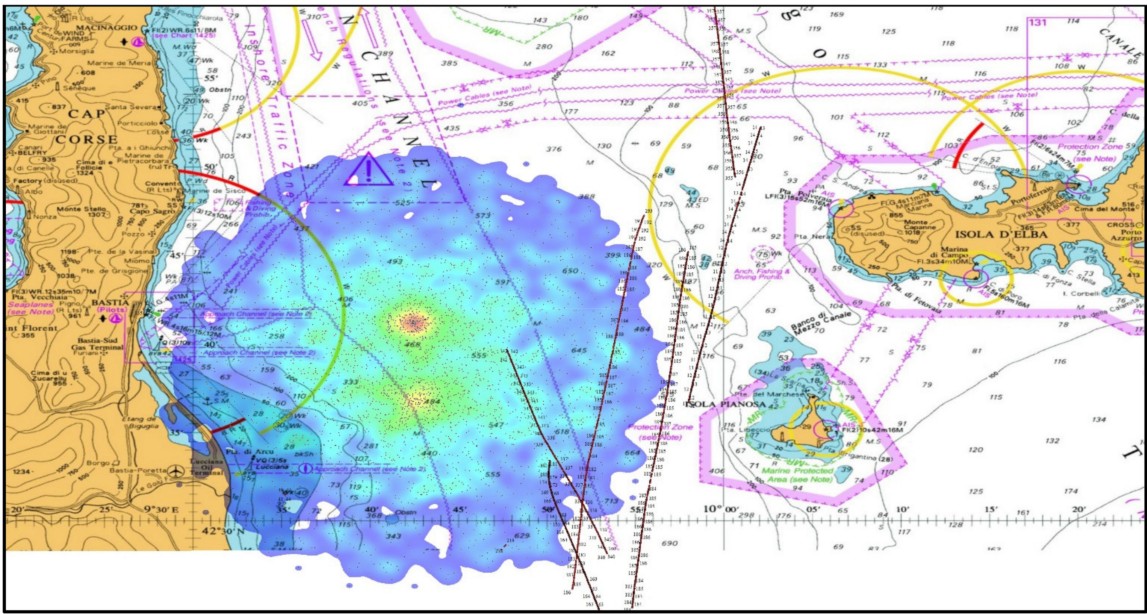

**Figure 10.** AIS spoofing—shipping density near Elba Island (Screenshot of ECDIS Navi Simulator, Wärtsilä) [40].

Considering the VDL load, the number of slots with received signal strength indicator (RSSI) above the noise level indicates that the spoofer has generated almost 100% VDL load. AIS message 4 (BS report) shows that the memory of the BS station is saturated, which means that the AIS BS in Monte Capanne has received more than 2048 different AIS station reports in the last 9 min.

It is difficult to find the position of the AIS generator because the signals were not transmitted synchronously with the AIS frame, so the time of arrival (TOA) information cannot be relied on to calculate the distance to the receiving station [51].

The first potential candidate is the vessel identified with the MMSI 999999999. The figure shows these MMSI trajectories over several months, with the vessel being most frequently in and off the main naval base in the area, as presented in Figure 11. It is very likely that the ship from the naval base is a different one from that received near the island of Elba, which in turn may be the same as the one received again near Bonifacio a few hours later. This might indicate the probability of several naval vessels sharing the same artificially generated MMSI, which is not unusual for warship AIS devices. Warship systems can generate such a spoofing event.

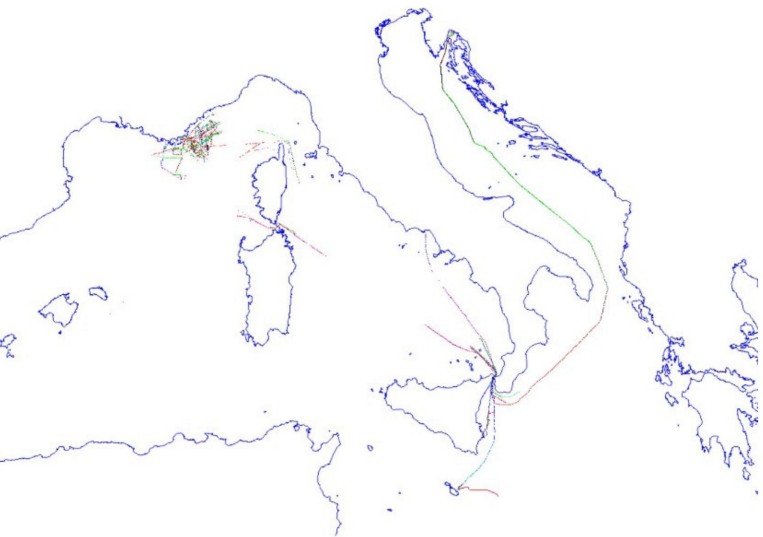

**Figure 11.** Three-month ship trajectory of MMSI 999999999.

Ships with MMSI 312320000 and MMSI 367309390 are also suspected. It is noticeable that the jitter error and RSSI level of these two targets match those of the spoofed messages with the Dutch MMSIs, as well as MMSI 999999999. This may indicate a naval ship conducting an electronic warfare exercise using two MMSIs. However, two transmitters sending AIS with the same power can be received with the same RSSI, even if they are in entirely different positions, i.e., because they can be more or less the same distance from the receiver BS. However, it should be noted that it is highly likely that warships in such activities wish to keep a shallow profile due to the particular sensitivity of these activities. Of course, we cannot rule out the possibility that fishing vessels performed the spoofing; but usually, they prefer to avoid attention rather than attract it. Also, they tend to use simpler manipulation methods on the AIS, though they are becoming increasingly adept at manipulating.

When looking at the received signal analysis (Figure 12), rapid changes are visible between one transmission and the other, which may indicate that the transmitting and receiving AIS stations are not in the line of sight and are therefore subject to signal fading: a ship at sea rolling on the waves may contribute. There is a large and abrupt drop at the end of the first burst that we do not quite understand. It could be a blind sector for the BS, or some sort of obstruction. There is a slight and constant decrease in the average RSSI from the middle to the end of the second burst, which could indicate that the distance between the transmitting and receiving AIS stations has increased, again indicating a moving spoofer.

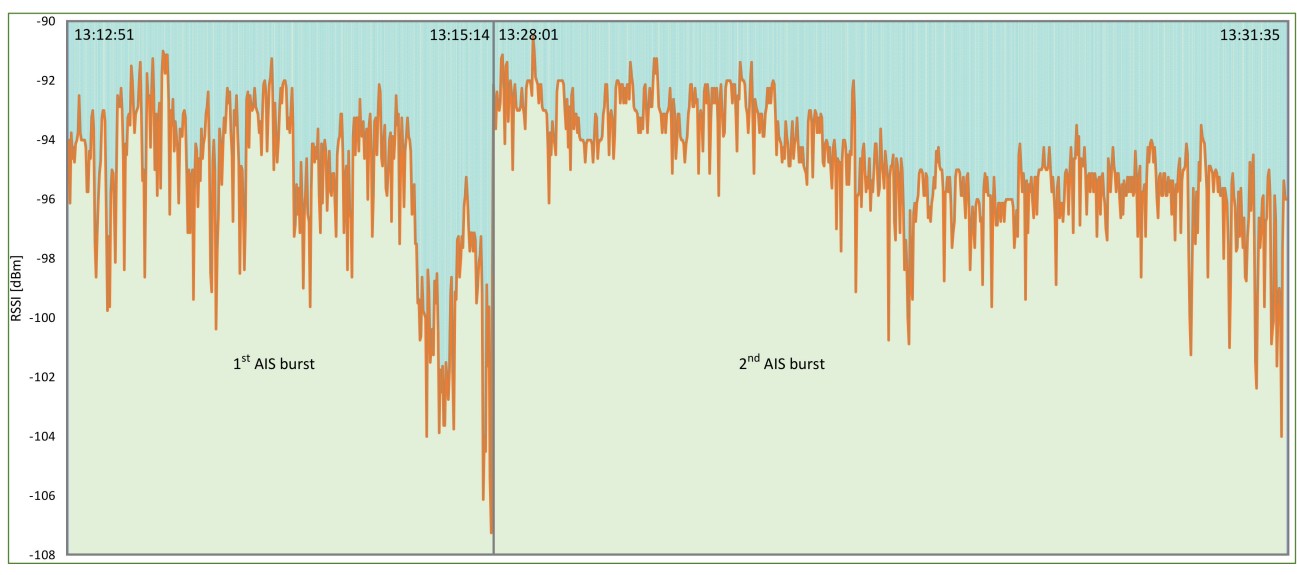

**Figure 12.** AIS RSSI signal during Elba Island spoofing event.

## 4.2. Potential Consequences of Spoofing Near ELBA Island

There were seven vessels found to be in a spoofing cloud, as shown in Table 5. To reinforce our research, a letter was sent to those in charge of the ISM of those vessels to enquire about the possible consequences for the safety of shipping. Even without their response, we were able, in our case study, to configure a navigation scenario in the affected area. The archive data were streamed using the application AIS Network Data Client and played in two different VTS applications Navi-Harbor (Wärtsilä) and Pelagus (Elman). The spoofing data was displayed on both ECDIS and RADAR applications via the ship tracking simulator Navi Trainer Pro (Wärtsilä).

**Table 5.** Ships involved in spoofing event near Elba on 3 December 2019.

| Ship Name | Type | IMO N° | MMSI | Gross Tonnage | Year of Build | Registered Owner |
|---|---|---|---|---|---|---|
| La Suprema | Ro-Ro | 9214288 | 247083700 | 49,257 | 2003 | Grandi Navi Veloci Spa (Italy) |
| Blue Star (ex. Minerva Alexandra) | Crude Oil Tanker | 9198094 | 273218590 | 58,125 | 2000 | Marine Trans Shipping Llc (Russia) |
| Eliana Marino | Ro-Ro | 9226360 | 247370400 | 18,265 | 2000 | Moby Spa (Italy) |
| Mega Express Five | Ro-Ro | 9035101 | 247183200 | 28,338 | 1993 | Forship Spa (Italy) |
| Leale | Chemical/Oil Tanker | 9404637 | 247238700 | 4831 | 2008 | Elbana Di Navigazione Spa (Italy) |
| Crudemed | Crude Oil Tanker | 9832547 | 636018645 | 62,330 | 2018 | Sans Souci Shipping Co (Liberia) |
| Cruise Ausonia | Ro-Ro | 9227429 | 247378700 | 30,907 | 2002 | Grimaldi Euromed Spa (Italy) |

For this case study, we have picked up m/v "Eliana Marino" within the area covered by the AIS spoofing cloud. We have synchronized the clock with the AIS data, and by using her actual position, course and speed, we were able to simulate what has been seen on her ECDIS screen just a minute before AIS spoofing. Figure 13 presents the information obtained by the radar (including AIS) overlay display on ECDIS. It can be seen that the navigation data are appropriate due to the matching of the ENC with radar and AIS data. This situation allows OOW to have an appropriate MSA for this particular navigational area.

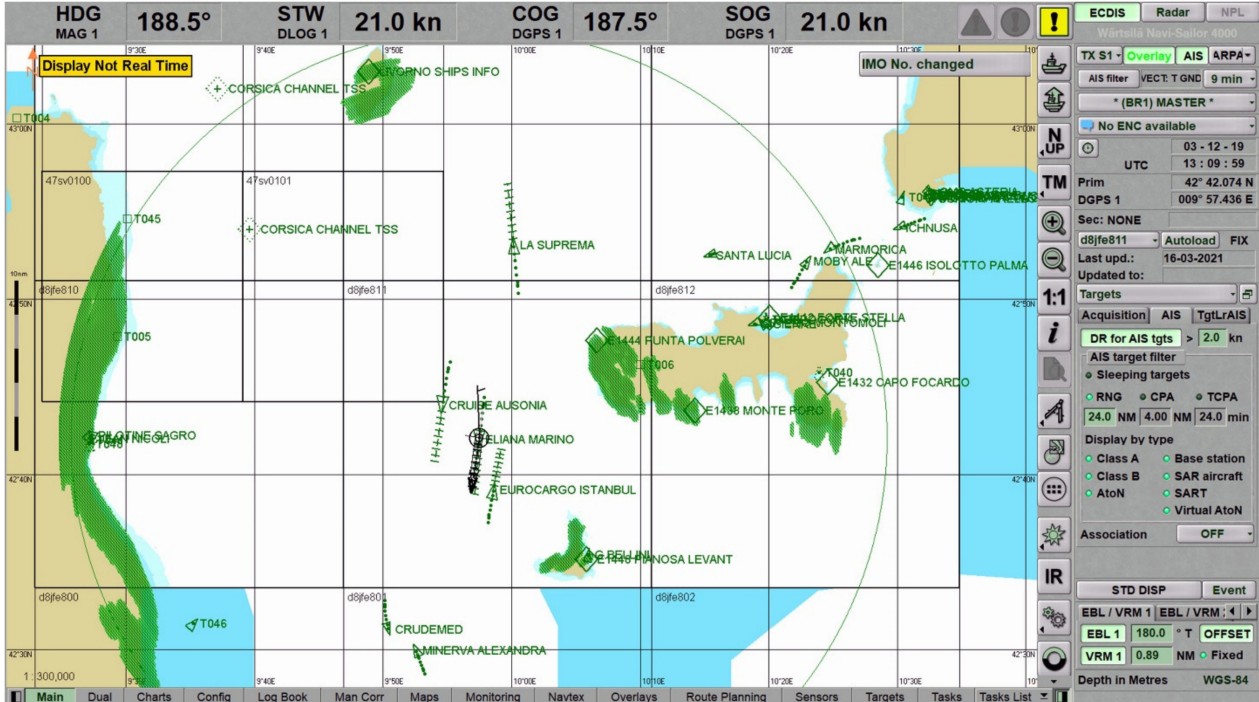

**Figure 13.** Simulation of the m/v "Eliana Marino" ECDIS display just a minute before the actual spoofing occurrence (Screenshot of ECDIS Navi Simulator, Wärtsilä).

Figure 14 presents the first burst of AIS spoofing event displayed on ECDIS with radar overlay and AIS data, in which significant degradation of MSA is caused, and 95% of the AIS signal processing load occurs.

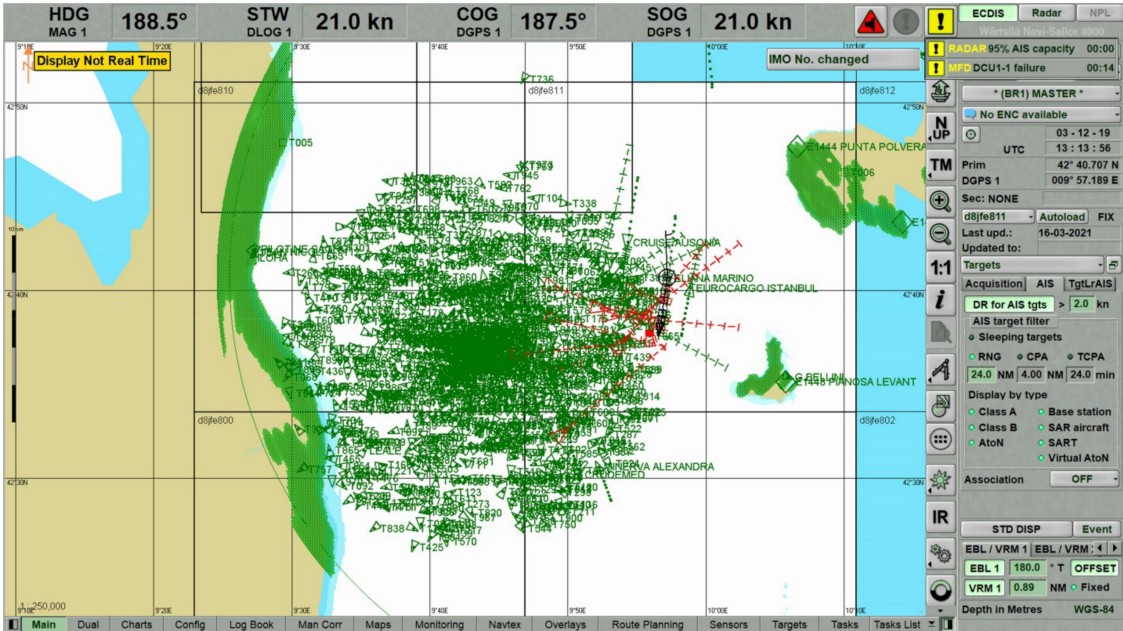

**Figure 14.** AIS processing overload (Screenshot of ECDIS Navi Simulator, Wärtsilä).

Consequently, the radar image shows a dangerous situation of m/v "Eliana Marino" on a collision course with more than a dozen other fake m/v, as shown in Figure 15.

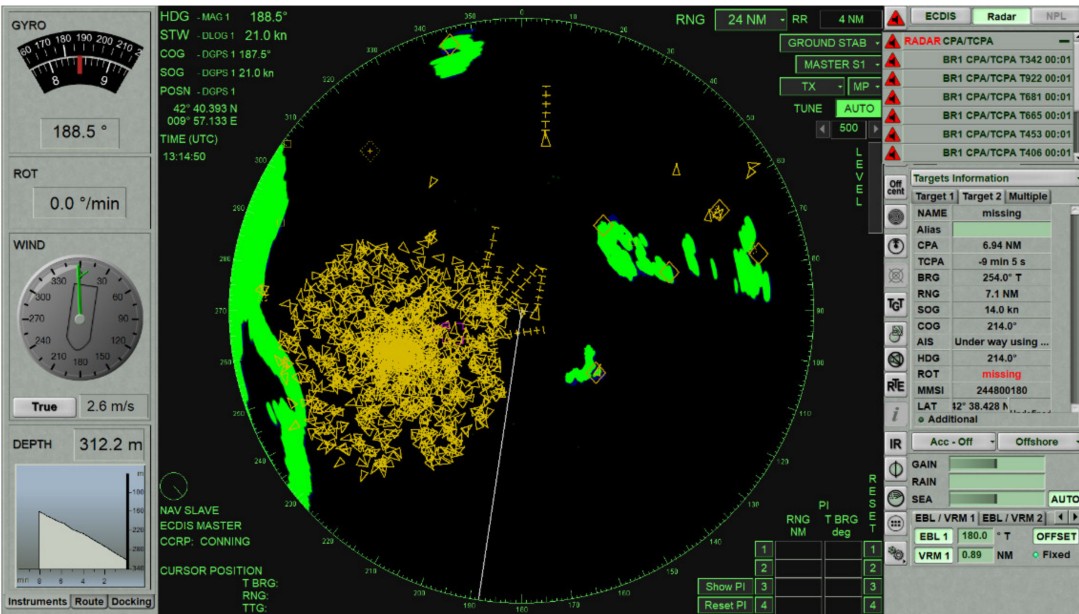

**Figure 15.** M/v "Eliana Marino" on a collision course with more than ten ships.

By filtering the presentation of sleeping AIS targets and limiting ships to a range within 6NM, the picture is slightly more apparent, as shown in Figure 16. The ECDIS/ARPA system is relieved, and it is also possible to change the CPA and Time to Closest Point of Approach (TCPA) filter. However, from the OOWs' perspective, this situation is hazardous in navigation since a relatively large number of false signals with collision course and a very short CPA are generated. Consequently, a relatively large number of collision alarms appear, which can further lead to inadequate OOW decisions. An experienced OOW will use in this situation a "raw" radar picture without the AIS data support and enhance sharp visual look-out. Luckily, the AIS spoofing event took place during the daytime and in a favorable navigation area. If it occurred during the night and in a dense maritime traffic area dangerous for navigation, it might have severe consequences on the safety of navigation. Therefore, the OOWs need to be aware of the possibility of AIS spoofing and the possible impact on the MSA.

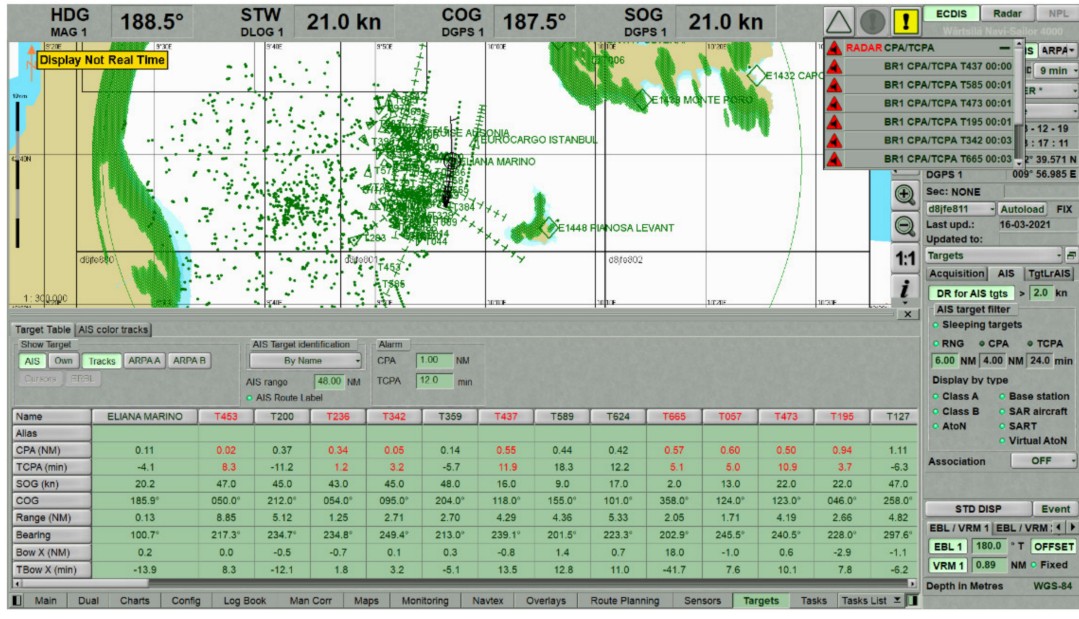

**Figure 16.** Filtering of AIS sleeping targets.

Figure 17 presents images of VTS system overload tested by two different applications Navi-Harbor (Wärtsilä) and Pelagus (Elman). It is evident that both systems are overloaded.

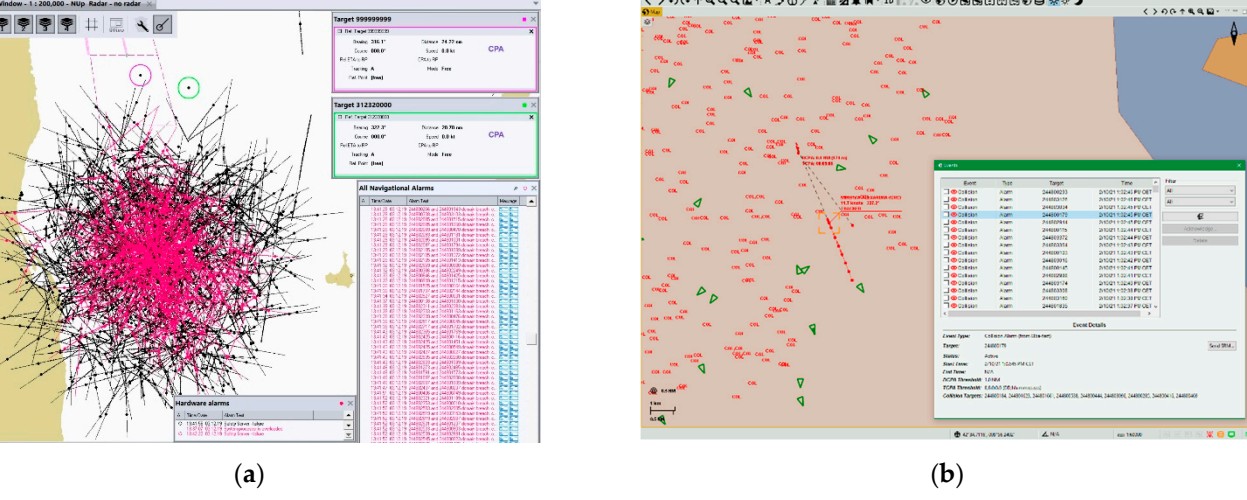

| (**a**) | (**b**) |

**Figure 17.** VTS applications—(**a**) Navi-harbor, (**b**) Pelagus.

## 5. Discussions

Along with the previous work by Iphar [35] and several other studies, our research has demonstrated AIS shortcomings. It is weakly secured and vulnerable to problems and attacks such as erroneous information, data corruption, and data spoofing. Prior to our case study of a specific spoofing event near Elba in December 2019, only a few cases in the Black Sea [38], the Strait of Hormuz [40], and the Port of Shanghai [42] were known to have a noticeable impact on maritime shipping.

In the Elba case study, we internalized the lesson that a spoofing situation in which a relatively large number of generated false signals with a collision course and a very short CPA can have severe consequences for the safety of navigation. We believe that the root cause of a successful AIS spoofing attack is not only that even the cheapest jammers available online can cause complete outages of the receiver signal [52], but a lack of appropriate technical security controls. The AIS user may mistakenly change course to avoid a collision with a non-existent ship, which could have severe consequences for the safety of navigation if it happens in a dense shipping area or where natural obstacles are near. In our case, the system also reached its maximum performance, revealing AIS saturation problems due to the high number of users. It is expected, paradoxically, that this problem may increase since IMO [53] increasingly aims to ensure that more ships are equipped with technology to ensure communication and improve safety.

From this perspective, VHF Data Exchange System (VDES) is seen as effective and efficient use of the radio spectrum, building on the capabilities of AIS to meet the increasing demands for data about the system [54]. Unlike AIS, VDES is much more robust and could benefit significantly from authentication mechanisms, such as digital (encrypted) signature techniques, which could be used to discard spoofed messages.

As we have already noted, AIS devices no longer follow today's technological developments and cyber threats. We believe that manufacturers should offer devices that provide the reliability and resilience that have previously passed the most demanding systematic deployment methods that an attacker would use to gain access, penetration testing [55]. Such testing should be performed by qualified testers using techniques such as simulating known risk assessments, attack scenarios, remote login solution security, social engineering, and testing of the physical perimeters that protect computer and network devices. It is preferable to perform penetration testing on production systems against test systems (which must be close duplicates of the targeted production systems to provide meaningful results) or against systems that are not yet in production, such as mandated

acceptance testing. In other words, testers must assess how AIS receivers respond to spoof signals and use this information to develop a counterstrategy to increase resistance to interference. There should be a set of standard tests that allow AIS users to select the best equipment for their application based on the degree of protection against jamming and spoofing. However, this type of testing can only be achieved through closer collaboration between the maritime sector, government security organizations, industry, and academia.

In other words, security requirements can be met with robust measures that take into account currently known threats that are taken into account during the production of new devices and systems. However, the history of technology and technique suggests that a sober review of the current situation requires an imaginative overview as well. No sane historian of nuclear weaponry would advise any stakeholder in a particular technology to be satisfied at any given time that their technology has reached a stable point and is invulnerable to attack. Techniques and technologies may be said to live lives of their own. Common sense dictates that nuclear devices be banned from the globe, yet of course, they are produced and sold to the point of super-redundancy. This is not all a matter of profiteering. We also cannot prevent techniques and technologies from capturing the imagination of some of the greatest scientific minds of our wealthier nations; so that now missile defense systems invite attacks that essentially prevent a non-scientific, non-technological process from eliminating such weapons. Hence the very nature of technique and technology devour the current, if not perpetually, unstable geopolitical and economic world. Rather than resolve a human problem through human means, humans are seduced into perpetuating their advantageous positions through technological advantages that most cannot afford. The very serious problem thus arises regarding the ethics of what some might call abusing technology (wealthy nations), and others might call mere opportunistic use of available techniques for striving to meet oppressors on an equal footing.

Specifically, in regard to this topic, there is little reason to believe that this process of securing means of trade will ever be relaxed; there is every reason to look at the circumstances of global geopolitics and economics realistically and therefore expect the process of security followed by insecurity and the need for advances in technique and technology to continue.

## 6. Conclusions

AIS represents a valuable tool for building MSA and enhancing the safety of navigation. It has numerous vulnerabilities and pitfalls as it is an open-sourced system transmitting on dedicated VHF frequencies. This feature is simultaneously the main advantage and disadvantage of the system. The simplicity of the AIS protocol has led to a wide variety of applications today, while AIS still lacks capabilities to verify the integrity and authentication of messages. From the navigators' point of view, it is of great importance to verify the information from AIS in navigation using radar and visual look-out. The essential problem for navigation safety is AtoN spoofing, especially for V-AtoNs. The integrity of V-AtoN messages is not so easy to verify. Therefore, navigators should pay special attention to this problem and use any available information sources to verify V-AtoN messages. It is easy for experienced navigators to detect some types of AIS spoofing, such as abnormal trajectories of AIS targets or a large number of spoofed signals, when they compare them with information from other sources. Navigators should never rely on a single source of information and should double-check the data provided by AIS.

A serious geopolitical consideration is that in an unequal economic world that is savagely competitive, certain nations will, when justifiably finding various maritime interference unfair, use vast resources to bypass current protocols. Scientists working to make AIS spoof-free are up against such countries and their allies, who feel no compunction in regard to violating rules and sanctions that they consider blatantly unjust, promoting inequity, and thus devote vast resources to evading control mechanisms.

In summary, we have pointed out the importance of cybersecurity. GNSS spoofing has been a defense issue for many years and is now beginning to affect shipping. As more

devices and autonomous systems rely on GNSS, even more systems could be vulnerable to spoofing attacks. The maritime industry and shipping are not immune to such cyberattacks, nor is the situation expected to improve soon. With hackers constantly looking for new ways to spoof and exploit AIS vulnerabilities, there will be a number of new cybersecurity openings in the future through which systems will be attacked if they are not adequately protected. Our analysis has shown that even a relatively easy to obtain AIS spoofing generator, such as that found near Elba Island, can have an impact on ship security. It was clearly pointed out that such a large number of ships appearing on the screen is primarily a technical problem, clearly projecting a false scenario. Under this mass of data, a ship can be missed, making it essential to simultaneously use other means of safe navigation. At a time when both AIS and GNSS, on which precise positioning is based, are subject to spoofing and denial-of-service attacks that threaten to render AtoN useless or even pose a threat to navigation, it may be unsafe to rely solely on ECDIS and its additional overlays.

Given the impact of digital technologies and to maintain seaworthiness, robust cybersecurity frameworks are needed. Disruptions to GNSS-based positioning and navigation have become a global phenomenon. To address a global problem, the GNSS community needs a global solution. The maritime industry needs to be ahead of the game, so manufacturers need to ensure the reliability, resilience, and functioning of multi-sensor systems for safety and liability reasons. For such a purpose, penetration testing is recommended that systematically employs methods that an attacker would use to gain access. The systems and sensors should be tested against test systems (that shall be close duplicates of the targeted in-production systems to provide meaningful results) or against systems not yet in production [55]. GNSS signals are essential for safe and efficient navigation. They are an integral part of maritime navigation. Degradation of these signals jeopardizes maritime safety.

This paper identifies the AIS vulnerabilities that impact maritime security and recommends that the maritime community implement a robust cybersecurity framework, use multiple GNSS constellations and encrypted signals that increase protection against spoofing and other maritime cyber threats. GNSS is not easily vulnerable, but we must be wary of those who say the risks are overstated.

**Author Contributions:** Conceptualization, A.A., I.P., J.M. and M.P.; methodology, A.A. and M.P.; data collection, A.A., I.P., J.M. and M.P.; validation, A.A. and M.P.; formal analysis, A.A., M.P. and I.P.; investigation, A.A. and M.P.; data curation, I.P., J.M. and M.P.; visualization, A.A. and M.P.; supervision, M.P.; writing—original draft preparation, A.A., I.P., J.M. and M.P.; internal review, M.P. All authors have read and agreed to the published version of the manuscript.

**Funding:** The publication of the paper is partially financed by the research project (L7-1847; Developing a sustainable model for the growth of the "green port") and the research group (P2-0394; Modelling and simulations in traffic and maritime engineering) at the Faculty of Maritime Studies and Transport, financed by the Slovenian National Research Agency.

**Institutional Review Board Statement:** Not applicable.

**Informed Consent Statement:** Not applicable.

**Data Availability Statement:** The data analyzed in this study was a reanalysis of existing data from the Mediterranean AIS Regional Exchange System (MARE∑) stored at the Slovenian Maritime Administration.

**Acknowledgments:** We thank the Italian Coastguard for providing the auxiliary sets AIS, which contain the VHF signal information (VSI) with signal strength (SS) and signal-to-noise ratio (SNR) at the receiving station. We thank Elman Srl and Wärtsilä Vessel Traffic Services.

**Conflicts of Interest:** The authors declare no conflict of interest.

## Abbreviations

| | |
|---|---|
| AIS | Automatic Identification System |
| AIS BS | AIS base station |
| AIS SART | Automatic Identification System Search and Rescue Transmitter |
| ARPA | Automatic Radar Plotting Aid |
| AtoN | Aid to navigation |
| CPA | Closest Point of Approach |
| EBIOS | Expression des Besoins et Identification des Objectifs de Sécurité |
| ECDIS | Electronic Chart Display Information System |
| ECS | Electronic Chart Systems |
| EEZ | Exclusive Economic Zone |
| EMODnet | European Marine Observation and Data Network |
| EPIRB-AIS | Emergency Positioning Indicating Radio Beacon-AIS enabled |
| FATDMA | Fixed Access Time Division Multiple Access |
| GMSK | Gaussian Minimum Shift Keying |
| GNSS | Global Navigation Satellite System |
| GPS | Global Positioning System |
| GT | Gross tonnage |
| IALA | International Association of Marine Aids to Navigation and Lighthouse Authorities |
| IEC | International Electrotechnical Commission |
| IMO | International Maritime Organization |
| ISM | International Safety Management |
| ITU | International Telecommunication Union |
| mIBC | Maritime certificate-less identity-based public key cryptography |
| MMSI | Maritime Mobile Service Identity |
| MOB-AIS | Man Over Board-AIS device |
| MSA | Maritime situational awareness |
| M/v | Motor vessel |
| NM | Nautical miles |
| NMEA | National Maritime Electronic Association |
| OOW | Officer of the watch |
| pAIS | Protected AIS |
| RF | Radio Frequency |
| RSSI | Received signal strength indicator |
| S-AIS | Satellite-based AIS |
| SAR | Search and Rescue |
| SEG | SafeSeaNet Ecosystem Graphical Interface |
| SOTDMA | Self-organized time division multiple access |
| SOLAS | Safety Of Life At Sea |
| SWOT | Strengths, Weaknesses, Opportunities, and Threats |
| TCPA | Time to Closest Point of Approach |
| TDM | Traffic Density Map |
| TESLA | Timed Efficient Stream Loss-tolerant Authentication |
| TOA | Time of arrival |
| V-AtoN | Virtual AtoN |
| VDL | VHF data link |
| VDES | VHF Data Exchange System |
| VHF | Very high frequency |
| VTS | Vessel Traffic System |

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
