# Peer review of "AIS Data Vulnerability Indicated by a Spoofing Case-Study"

_applsci, doi:10.3390/app11115015_

Round 1

Reviewer 1 Report

This paper describes the necessity of technologies to protect from cyberattacks. The analytical result of AIS spoofing case where a large number of fake AIS messages is shown. The importance of the OOWs skill to check multi source of information is also mentioned. The reviewer agrees the authors opinion. 

This paper only summarizes the current situation and the requirement of cybersecurity framework. The reviewer thinks that many of manufacturers already know the importance of the cybersecurity framework. Manufacturers don't know how to ensure the reliability, resilience and functioning of multi-sensor systems for safety and liability reasons. The authors recommend the penetration testing. But manufacturers cannot understand what kinds of penetration testing is enough to ensure the cybersecurity. Do the authors have any concrete idea to carried out penetration testing for maritime navigation systems?

Author Response

Dear Reviewer,

Thank you very much, indeed, for your valuable comments. We have tried to take your suggestion into account in the best possible way to add more value to the article. We sincerely hope that we managed to answer your respective questions in the new Chapter 5 (article attached).

Yours Sincerely,

Andrej Androjna (et.al.)

Reviewer 2 Report

The paper investigates the security of the Automatic Identification System (AIS). In particular, the authors carry out a literature review on the implications of AIS spoofing and analyse data collected from a specific security incident. They conclude by pointing out the need of a cyber security framework for maritime systems.

To topic is current and interesting.

The paper is original in terms of presenting and analysing data that were collected from various sources such as the Italian coastguard and the Slovenian Maritime administration about a security incident.

The literature review seems not so well focused. Although the authors mention that the focus is on ‘what are the implications of AIS spoofing in the maritime domain’, they go on and present works on AIS vulnerabilities and security controls for mitigation, which doesn’t directly answer the question that was presented before.

Moreover, the purpose of the SWOT analysis, presented in the same section is not clear and add more confusion. The authors should explain why do they need to present the strengths and opportunities of AIS service at this stage.

In terms of research contribution, it seems to be limited at the moment. Nevertheless, the paper can be enhanced by adding a discussion section before the conclusion. What can be discussed are the causes of successful AIS spoofing attacks. Is it lack of security policies in the maritime sector or is it lack of suitable technical security controls? Can spoofing attacks be prevented with existing security controls, such as the ones presented in the literature review?  From a cybersecurity perspective these are some interesting questions and it would be nice if the authors could derive some answers from the analysis of the incident that they have carried out.

typo line 591  ‘pro-vided’

Author Response

Dear Reviewer,

Thank you very much, indeed, for your valuable comments. We have tried to take your suggestion into account in the best possible way to add more value to the article. We sincerely hope that we managed to answer all your respective questions (new article attached).  

Changes/modifications are as follow:

  1. Regarding the literature review, we agree with your corresponding comment; however, in this paper, we mainly focused on AIS spoofing and also mentioned some other AIS vulnerabilities, as we already mentioned in our previous research study (J. Mar. Sci. Eng. 2020, 8(10), 776; https://doi.org/10.3390/jmse8100776). We have also considered some additional available literature/references (nos. 15, 35, 52, 53 and 54) on the mentioned topic.
  2. The analysis SWOT has been deleted regarding the recommendations.
  3. We have tried to consider your suggestion in the best possible way by adding Chapter 5 - Discussion. Sincerely, we can easily see the benefit from it. Thank you for your recommendation.
  4. Inspired by the comment (typo line 591 'provided), we have tried to add some more 'spice' to Chapter 5.

Yours Sincerely,

             Andrej Androjna (et.al.)

Reviewer 3 Report

The paper presents the results of the analysis of the AIS spoofing incident near Elba in December 2019. This is complemented by a review of literature on the security of the AIS. 

The paper addresses an important problem, namely the security of the AIS. The methodology used to analyze the specific incident is sound, and the results are interesting and provide useful insight. 

The paper is well structured and clearly written, and can be easily followed even by readers not necessarily familiar with the AIS. 

However, some weaknesses also exist:

  1. As AIS spoofing is closely related to GPS spoofing, the keywords "AIS" and "spoofing" that were used for searching may not have captured all the relevant literature. Adding "GPS", and possibly "GNSS" to the keyword set may provide additional relevant sources.
  2. The title of reference [26] (which does not seem to be publicly available, please provide a url) suggests that overlap may exist between the material in [26] and the paper under review. The added value of the paper over [26] needs to be clearly delineated.
  3. The SWOT analysis, although useful for tabulating the ups and downs of the AIS service, is not the most appropriate tool to employ in the context of the paper. A risk analysis of the service would be much more appropriate, even if only to highlight the involved risks.
  4. Three proposals for mitigating AIS-related risks are discussed, namely references [29], [9], and [16, 36]. An extended version of [9] is available in: S. Sciancalepore, P. Tedeschi, A. Aziz and R. Di Pietro, "Auth-AIS: Secure, Flexible, and Backward-Compatible Authentication of Vessels AIS Broadcasts," in IEEE Transactions on Dependable and Secure Computing, doi: 10.1109/TDSC.2021.3069428. Several other solutions have been proposed in the past, some of which are reviewed in this reference, as well as in [29] and [16, 36]. All three of the methods in [9], [29], and [16,36] use cryptographic techniques; they assume an underlying infrastructure; and they exploit messages (6 and/or 8) of the AIS protocol, without modifying the protocol itself. Hence, all three methods are backward-compatible, and they do not require any changes in the AIS service; they do however require changes in software/firmware/hardware of the AIS devices. The method in [9] provides authentication service only, whereas that in [29] provides authentication and message integrity service, and [16,36] provides authentication, message integrity, and confidentiality services. The text on p.7 should then be accordingly modified.
  5. Some discussion on the potential of VDES to mitigate spoofing attacks would provide additional insight.    

Author Response

Dear Reviewer,
Thank you for your encouraging comments. We have tried to take your suggestions into account as best we can to enhance the article. We sincerely hope that we have succeeded in answering all your respective questions (new article attached). 
The changes/amendments are as follows:
1. The recommended article has been considered: S. Sciancalepore, P. Tedeschi, A. Aziz, and R. Di Pietro, "Auth- AIS: Secure, Flexible, and Backward-Compatible Authentication of Vessels AIS Broadcasts," in IEEE Transactions on Dependable and Secure Computing, doi: 10.1109/TDSC.2021.3069428

2. The reference (26. Perkovič, M. AIS spoofing near Elba Island analysis and research data; University of Ljubljana, Faculty of Maritime studies and transport: Ljubljana, Slovenia; 2020) has been deleted as we do not have a doi and it is not publicly available.
3. We agree with your comment regarding the selection of our keywords. however, in this paper we have primarily focused on AIS spoofing and also mentioned some other AIS vulnerabilities, as we already presented results on GPS /GNSS vulnerabilities in our previous research study (J. Mar. Sci. Eng. 2020, 8(10), 776; https://doi.org/10.3390/jmse8100776). We followed your hint and searched the available literature again and added some recent references (nos. 15, 35, 52, 53, and 54) on the mentioned topic.
4. The analysis SWOT has been deleted in relation to your and the Reviewer 2 recommendations. AIS recommendations in relation to risks have been highlighted.

5. in relation to the mitigation of AIS -related risks, the text has been amended and updated to reflect the recommendation. The text between lines 288 and 308 is now clearly written and the capabilities of the proposals of the cited authors have been highlighted. The suggested articles (Sciancalepore et al.) have also been analyzed and cited as a valuable source.
6. We have attempted to address your suggestion about the potential of VDES to mitigate spoofing attacks in Chapter 5. Frankly, we can easily see the benefit from it. We thank you for your recommendation.

Sincerely, Andrej Androjna (et.al.)

Round 2

Reviewer 2 Report

Thanks for the updated version of the paper.

The discussion section enhances the value of the paper.